# Engineering a riboswitch-based genetic platform for the self-directed evolution of acid-tolerant phenotypes

Hoang Long Pham[1,2], Adison Wong[1,2,3], Niying Chua[1,2], Wei Suong Teo[1,2,4], Wen Shan Yew[1,2] & Matthew Wook Chang[1,2]

Environmental pH is a fundamental signal continuously directing the metabolism and behavior of living cells. Programming the precise cellular response toward environmental pH is, therefore, crucial for engineering cells for increasingly sophisticated functions. Herein, we engineer a set of riboswitch-based pH-sensing genetic devices to enable the control of gene expression according to differential environmental pH. We next develop a digital pH-sensing system to utilize the analogue-sensing behavior of these devices for high-resolution recording of host cell exposure to discrete external pH levels. The application of this digital pH-sensing system is demonstrated in a genetic program that autonomously regulated the evolutionary engineering of host cells for improved tolerance to a broad spectrum of organic acids, a valuable phenotype for metabolic engineering and bioremediation applications.

---

[1] Department of Biochemistry, Yong Loo Lin School of Medicine, National University of Singapore, Singapore 117596, Singapore. [2] NUS Synthetic Biology for Clinical and Technological Innovation (SynCTI), Life Sciences Institute, National University of Singapore, Singapore 117456, Singapore. [3] Present address: Singapore Institute of Technology, 10 Dover Drive, Singapore 138683, Singapore. [4] Present address: School of Life Sciences and Chemical Technology, Ngee Ann Polytechnic, 535 Clementi Road, Singapore 599489, Singapore. Hoang Long Pham and Adison Wong contributed equally to this work. Correspondence and requests for materials should be addressed to M.W.C. (email: matthew_chang@nuhs.edu.sg)

Cellular metabolism and growth are dynamically regulated by fundamental environmental signals such as light, temperature, and pH. The ability to program complex cellular behavior in response to specific environmental cues can potentially lead to interesting applications in basic research, healthcare, and biomanufacturing[1–4]. For example, light-responsive genetic programs have been realized based on engineered photosensitive DNA-binding proteins[5–8]. These optogenetic tools permit temporal and spatial control of gene expression, hence enabling the execution of complex biological functions, such as bacterial photography or edge-detection[9, 10]. Similarly, heat-regulating machines comprising either thermosensitive proteins or RNA motifs have been developed for applications in diagnostics, biocontainment, and biomanufacturing[11–15].

The programming of pH-homeostasis in living cells has yet to achieve comparable success. It is only recently that scientists have successfully demonstrated the capacity to implement pH sensing and control mechanisms in mammalian chassis. Several interesting biotechnological applications have emerged in the process, including a gas-inducible gene expression control system for industrial biomanufacturing and a prosthetic genetic program that corrects diabetic ketoacidosis in living animals[4]. Synthetic acid-inducible promoters in yeast have also been used to regulate organic acid production, leading to a tenfold improvement in lactic acid production under low-pH fermentation conditions compared to the use of the standard constitutive promoter TEF1

for gene expression[16]. However, similar tools in bacterial chassis remain scarce and the lack of a versatile pH-sensing toolbox has limited progress in tinkering with biological systems for pH-related applications, such as creating designer probiotics for in vivo diagnosis and treatment of acid reflux[17]. The molecular mechanisms of pH homeostasis in bacteria have been elucidated, but their complexity poses a paramount challenge for genetic component mining. For instance, at least 11 regulatory proteins at various levels of the stress response signaling cascade are required to induce a glutamate-dependent protective response against acidic challenge (pH 2.5) in E. coli[18–23]. In addition, pH-inducible expression systems isolated from Helicobacter pylori[24], Lactococcus lactis[25], and Lactobacillus acidophilus[26] can only be applied in their native hosts due to the requirement of organism-specific factors for activation.

In this study, we report the development of a set of riboswitch-based pH sensors enabling the precise control of bacterial gene expression in response to differential environmental pH conditions. Our engineered riboswitch can digitalize pH-dependent gene expression for high-resolution memory-based detection of external pH with an OFF-ON output difference of up to 31-fold. To further illustrate the practicality of the engineered riboswitch, we interlink pH sensing with the error-prone DNA replication machinery and fluorescent cell labeling, thus programming E. coli for autonomous evolutionary engineering and enrichment of the acid-tolerant phenotype.

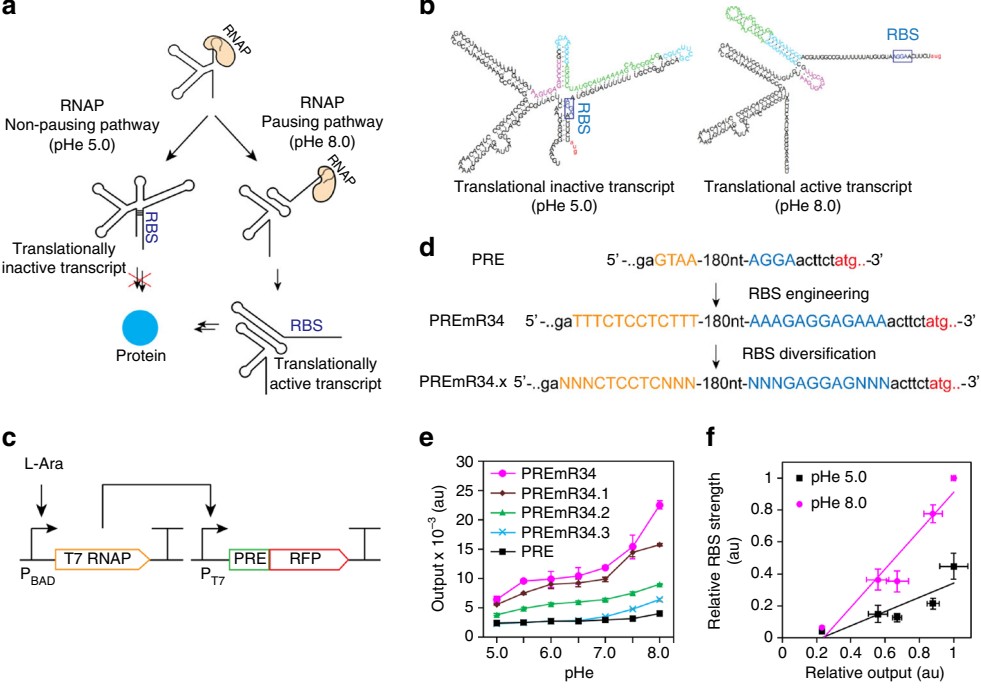

**Fig. 1** Engineering the dynamic range of a wild-type pH-riboswitch. **a** Mechanism of action of the wild-type pH-riboswitch. During transcription, the wild-type pH-riboswitch adopts distinct folding conformations to affect the RNAP-dependent mRNA synthesis process. At low pH (pHe 5.0), mature mRNA is produced via the RNAP non-pausing pathway to yield translationally inactive transcripts with the ribosome-binding site locked by its complementary sequence. At high pH (pHe 8.0), mRNA is produced via the RNAP pausing pathway to yield translationally active transcripts with an accessible ribosome-binding site to allow translation. **b** Detailed folding conformations of inactive (*left*) and active (*right*) states of the wild-type pH-riboswitch. The putative wild-type ribosome-binding sequence (AGGA) is *boxed*. **c** Schematic of the genetic construct used to characterize the dynamic range of the wild-type and synthetic pH-riboswitches. **d** Engineering strategy to tune the dynamic range of the wild-type pH-riboswitch. The weak ribosome-binding site and its complementary sequence within the wild-type pH-riboswitch (PRE) were replaced by those of a strong ribosome-binding site RBS34 (BBa_0034) to generate the synthetic variant PREmR34. A library of PREmR34 variants (PREmR34.x) with diverse dynamic ranges was generated by varying the bases surrounding the "core" RBS sequence of PREmR34 (GAGGAG). **e** Output fluorescence at varied extracellular pH (pHe) of the pH-sensing genetic devices (L-ara 0.02% m/v). **f** Correlation between the fluorescence output and relative strength of the RBSs embedded in the pH-riboswitch variants. The relative RBS strengths were normalized against RBS34, and the relative outputs were normalized against the output of the pH-sensing device built from RBS34. The data represent the mean of three independent experiments performed on different days

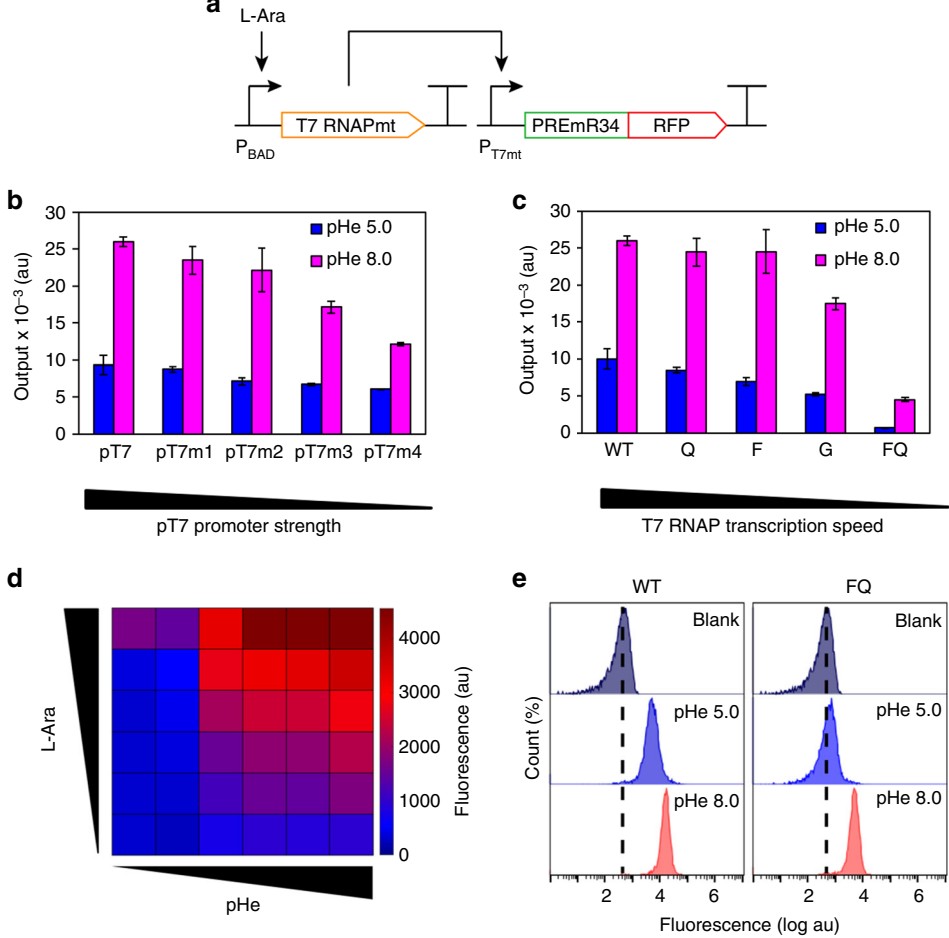

**Fig. 2** Alleviation of the background leakiness of the pH-sensing device through transducer engineering. **a** Schematic of transducer engineering strategy involving the generation of T7 promoter mutants (pT7mt) and T7RNAP mutants (T7 RNAPmt). **b** Tuning the basal expression of the pH-sensing device through T7 promoter engineering. **c** Tuning the basal expression of the pH-sensing device through T7RNAP engineering. **d** Tuning the basal expression of the pH-sensing device through titration of the intracellular T7RNAP concentration. The heat map shows the graded fluorescence output of the pH-sensing device controlled by the mutant FQ under varying L-arabinose induction (bottom to top: 0, 0.001, 0.002, 0.008, 0.02, 0.1% m/v) and extracellular pH (pHe) conditions (*left* to *right*: 5.5, 6.0, 6.5, 7.0, 7.5, 8.0). **e** Comparing the output of the pH-sensing devices controlled by wild-type T7RNAP and the mutant FQ at L-arabinose induction 0.002% m/v. The data represent the mean of three independent experiments performed on different days

We envision that the riboswitch design principles and experimental framework presented herein can be broadly applied to develop valuable phenotypes for industrial biotechnology and bioremediation applications.

## Results

**Engineering a wild-type pH-riboswitch**. The discovery of a 207-nucleotide RNA element that regulates the expression of the *alx* gene in *E. coli* in a pH-dependent fashion was previously reported[27, 28]. This wild-type pH-responsive RNA element (PRE) functions by adopting distinct folding conformations co-transcriptionally to affect mRNA synthesis (Fig. 1a). Under extracellular pH (pHe) 6.8, the PRE forms an inactive structure that allows non-pausing transcription to yield mature translationally inactive transcripts (OFF structure) with a ribosome-binding site (RBS) sequestered by its complementary sequence. Under pHe 8.0, the formation of stem loops within the PRE induces transcriptional pausing, which leads to the formation of mature translationally active transcripts with an RBS accessible for translation (ON structure) (Fig. 1b). To examine the pH-sensing performance of the PRE, we constructed a 2-plasmid genetic device consisting of a low-copy (SC101 origin) plasmid

encoding T7 RNA polymerase (T7RNAP) under the control of a $P_{BAD}$ promoter and a high-copy (ColE1 origin) plasmid encoding red fluorescent protein (RFP) fused downstream of the PRE and T7RNAP cognate promoter (Fig. 1c). PRE is the sensing element that detects changes in pH, whereas the T7RNAP-pT7 pair functions as the transducing element to process the input signal into a measurable RFP output as the actuator. The minimum and maximum induction levels are defined at pHe 5.0 and 8.0 due to the limited growth capacity of the *E. coli* host outside this pHe range. Following induction with L-arabinose and titration across differential pHe values, the wild-type riboswitch elicited only a narrow ~ 2.5-fold activation of RFP expression between pHe 7.0 and pHe 8.0, with an output amplitude ranging from 1000 to 2400 relative fluorescence units per cell (rfu) (Fig. 1e).

We hypothesized that the limited dynamic range of the wild-type PRE is attributable to its weak putative RBS sequence (AGGA), which exhibited low steady-state expression when evaluated against other synthetic RBSs in an independent RBS strength comparison assay (Supplementary Fig. 1a). To improve the dynamic range of the PRE, we replaced the weak 4-nucleotide wild-type RBS sequence with a strong 12-nucleotide synthetic RBS34 (BBa_0034) to create a synthetic PRE variant, PREmR34 (Fig. 1d). The corresponding 12-nucleotide complementary

sequence near the 5′-end was designed to fully pair with RBS34 in the OFF structure. Computational modeling using mfold demonstrated that the OFF structure of PREmR34 is similar to that of PRE, suggesting that the modifications at the RBS sequence did not affect the folding structures of the PRE. This similarity is consistent with our desire to engineer the dynamic range while conserving the natural pH-sensing behavior of this riboswitch (Supplementary Fig. 1b)[29]. The pH-sensing device built from PREmR34 detected values between pH 5.0 and pH 8.0 with a 3.5-fold activation ratio and an output amplitude ranging from 6400 to 22,500 rfu (Fig. 1e and Supplementary Fig. 1c).

We reasoned that the PREmR34 dynamic range could be adjusted by simple modifications in the RBS region. A RBS library was first constructed by degenerating the nucleotides (three upstream and three downstream) surrounding the 6-bp "core" sequence (GAGGAG) of the RBS34-RFP construct. RBS variants with differential translation efficiencies in the RBS strength measurement assay were then isolated, sequenced and used to construct new PREmR34 variants (Supplementary Fig. 1a). In each of these variants, complementary sequences near the 5′-end of the PRE were designed for full pairing with the 12-bp RBS sequence (Fig. 1d), allowing the translation rates of the embedded RBSs to be tuned without introducing drastic changes to the folding structures of the scaffold variant PREmR34 (Supplementary Fig. 1b). Using this strategy, we obtained a library of pH-sensing devices with a gradient dynamic range and output levels that were linearly correlated with the strength of the RBSs embedded within the engineered PREs, hence supporting a framework for the predictable amplification of the riboswitch outputs (Fig. 1e, f). Altogether, these results imply that RBS is a modular region within the PRE that can be exploited to customize its dynamic range and amplitude without affecting its pH-sensing behavior.

*E. coli* is neutralophilic in nature and may execute homeostatic mechanisms against drastic changes in pHe, thereby affecting the interpretation of the intracellular pH (pHi) by our engineered riboswitch. To evaluate the effect of pHe conditioning on pHi, we constitutively expressed *pHluorin2*, a ratiometric GFP variant with a pH-sensitive bimodal spectrum commonly used for live-cell pHi measurements[30]. The 410/470 excitation ratios of cells grown at various pHe values were recorded, calibrated, and used for pHi calculations (Supplementary Figs. 2 and 3). *Escherichia coli* TOP10 (T10) maintained a stable pHi in acidic media (pHe 5.0 to 7.0), and a steeper linear increase in pHi with respect to pHe was observed only in more alkaline media (pHe 7.0 to 8.0) (Supplementary Fig. 2d). Plotting the response curves of pH-sensing devices built from PRE and PREmR34 against pHi further illustrated this characteristic, as indicated by the sharp linear increase in the output expression in the alkaline pHi region between 7.2 and 8.3 (corresponding to pHe between 7.0 and 8.0; Supplementary Fig. 1 and Supplementary Fig. 5). In addition, the pHe in M9 glycerol T10 cultures was fairly constant in the absence of chemical buffers and did not vary by more than 0.35 pH units over 12 h (Supplementary Fig. 3). This observation may imply that wild-type T10 did not increase pHe through urease and arginine deiminase activities, in contrast to observations in other acid-tolerant bacteria species[31]. Furthermore, the effects of pHe on the well-known promoters used in this study, $P_{BAD}$ and J23119, were investigated (Supplementary Fig. 4). For both promoters, at pHe 5.0, a slight reduction in RFP expression was observed (~ 8% of maximal expression at pHe 7.0). Above this pHe level, RFP expression remained constant. Taking into consideration that the sensitive range of the pH-sensor is between pHe 6.0 to 8.0, the pH-dependent gene expression can be interpreted to be under the regulation of the riboswitch element.

**Alleviating background leakiness of pH-sensing device.** Leaky background expression is generally undesirable for genetic biosensors. Although our original pH-sensing device composed of wild-type T7RNAP and engineered PRE exhibited a broad dynamic range, considerable basal expression was observed in the OFF state at pHe 5.0. We postulated that the observed leakiness was due to the rapid transcription speed of the T7RNAP transducer and consequent generation of excessive mRNA transcripts from the marginal intracellular T7RNAP pool. Hence, two strategies were considered to improve the transducing element of the pH-sensing device (Fig. 2a): (i) engineering the T7 promoter to reduce the promoter strength of the T7RNAP-based expression system and (ii) engineering T7RNAP to modulate its inherent transcription speed. For promoter engineering, we employed a small library of T7 promoters previously derived for the fine-tuning of T7RNAP-based gene expression control in *E. coli*[32]. While we were able to tune the output expression of the pH-sensing device in the ON state at pHe 8.0, the approach was less effective in reducing the background expression in the OFF state (Fig. 2b). In particular, the weakest T7 promoter variant used in our biosensor still resulted in leaky gene expression at pHe 5.0 with minimal L-arabinose induction. T7RNAP engineering was next pursued to alleviate the background leakiness of the pH-sensing device. Previous studies have reported that T7RNAP can be customized for attenuated processivity through single amino-acid substitutions at active sites[33]. To determine if the use of T7RNAP with reduced transcription speed would reduce the background leakiness, we developed a library of T7RNAP mutants for insertion in the pH-sensing device built with PREmR34 in place of wild-type T7RNAP (Supplementary Fig. 6). Convincingly, the output expression of the pH-sensing device in both the ON and OFF states was lowered by reducing the polymerase processivity through single point mutations around the T7RNAP active sites (Fig. 2c). Exploiting this property additively by using a T7RNAP double mutant (F644AQ649S) further reduced the processivity and yielded a pH-sensing device with even tighter basal expression in the OFF state (output 1259 rfu) (Fig. 2c). Further optimization of the intracellular T7RNAP concentrations by controlling L-arabinose induction levels ultimately yielded a pH-sensing device with significantly reduced background expression in the OFF state (output 214 rfu) (Fig. 2c–e).

**Memory-based digitalization of pH-riboswitch.** The best engineered pH-riboswitch variant developed thus far (PREmR34) exhibited an ON/OFF activation ratio within one order of magnitude (3.5-fold). This activation ratio is potentially challenging for integration into higher-ordered circuits because small differential outputs could be easily obscured by noise and cell heterogeneity, resulting in biosensor malfunction. To exploit the unique pH-sensing capacity of the PRE despite its narrow activation ratio, we applied DNA memory elements to digitalize its response curve and amplify the output difference between the OFF (pHe 5.0) and ON (pHe 8.0) states. Central to this approach was the utilization of the pH-riboswitch to regulate the expression of recombinases and catalyze the unidirectional switching of the DNA promoter flanked by recombinase recognition sites. By incorporating DNA recombinase as an additional processor of the intermediary signal, our engineered biosensor could be implemented as a high-pass filter to enable DNA switching only at high input signals at which the intracellular integrase concentration exceeded a required threshold. Consequently, the activation ratio of the whole sensing device could be digitally amplified by several orders of magnitude depending on the strengths of the embedded constitutive promoters.

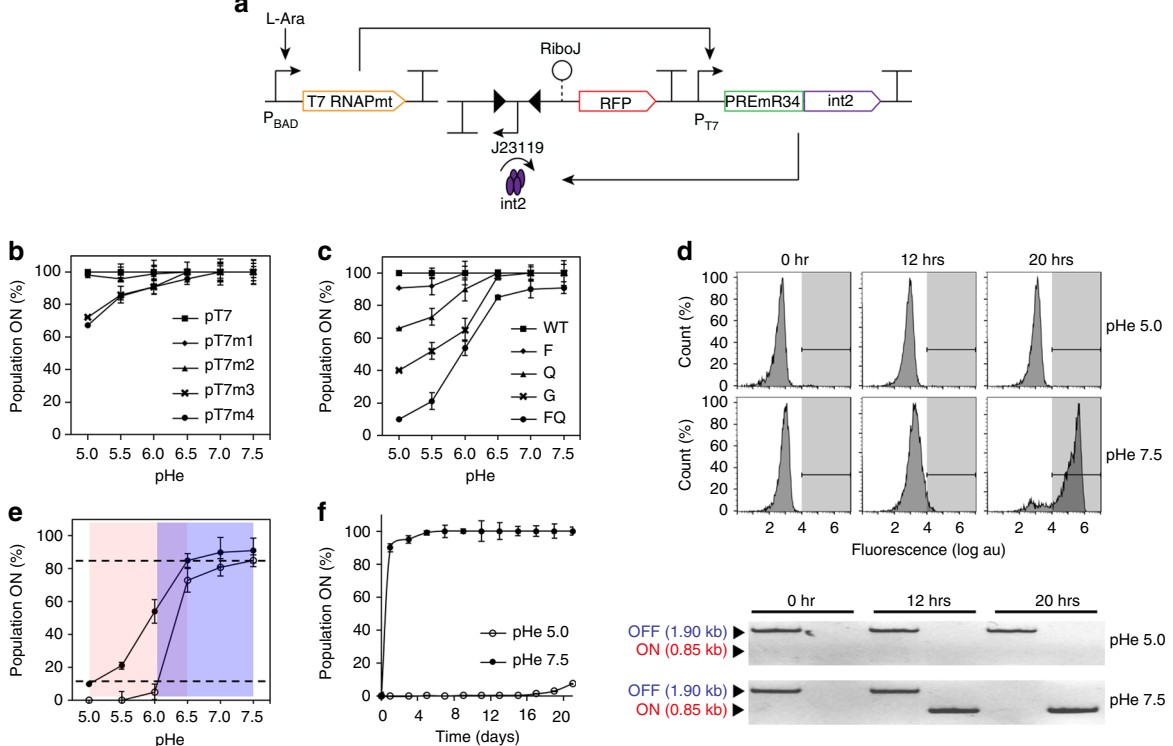

**Fig. 3** Characterization of the digitalized pH-sensing system. **a** Schematic of the digitalized pH-sensing system. The system regulates the pH-dependent expression of an integrase (int2) that catalyzes the unidirectional reorientation of a strong constitutive promoter (BBa_J23119) flanked by its recognition sequences. At acidic pH (pHe 5.0), integrase expression is suppressed, and cells are maintained in the "OFF" state (no fluorescence). At neutral pH (pHe 7.5), integrase expression leads to the reorientation of the promoter J23119 to drive RFP expression, and cells are switched to the "ON" state (*high fluorescence*). **b** Outputs of the digital pH-sensing systems constructed from the T7 promoter variants. **c** Outputs of the digitalized pH-sensing systems constructed from the T7RNAP variants. **d** Time-course characterization of the digitalized pH-sensing system. *Top*: The histograms show the fluorescence distribution of the cell population induced at pHe 5.0 or 7.5 over time. The *shaded* regions indicate the "ON" state (*red fluorescence* $> 10^{3.8}$ au). *Bottom*: Gel electrophoresis of PCR products corresponding to the "ON" (0.85 kb) and "OFF" (1.90 kb) states of the cell population induced at pHe 5.0 or 7.5 over time. **e** Tuning of the transition band of the digital pH-sensing system through titration of the intracellular T7RNAP concentration. Low-threshold (*filled circles*) and high-threshold (*open circles*) transition bands were obtained at near-saturated (0.1% m/v) and lower induction concentrations (0.002% m/v) of ʟ-arabinose, respectively. The *shaded regions* indicate the transition regions. The *dashed lines* indicate 10% and 85% population switching. **f** Stability assessment of the digitalized pH-sensing system at pHe 5.0 (*blue*) and pHe 7.5 (*red*) (with 0.002% m/v ʟ-arabinose) over long periods. The switching behavior of the system was monitored daily over a 21-day period. The data represent the mean of three independent experiments performed on different days

A library of 11 orthogonal serine-type phage integrases was identified from the literature as suitable components to implement our approach[34]. Previously, it was shown that the expression of these integrases above a critical threshold within host cells induces irreversible DNA switching with no cross-talk to the *E. coli* recombinase system. We selected five integrases (int2, int4, int5, int8, int9) from this library and compared their dynamic behavior in a 1-plasmid system within a configuration relevant to our downstream applications (Supplementary Fig. 7a). In agreement with previous studies, integrase 2 (int2) exhibited a wide dynamic range and high tolerance toward leaky expression levels (Supplementary Fig. 7b). This integrase was, therefore, used for further circuit construction. Next, a digital pH-sensing system was designed to utilize a core pH-sensing module to regulate the expression of int2, which controlled the physical orientation of the constitutive promoter J23119 (BBa_23119) embedded within int2 recognition sites (*attB* and *attP*) (Fig. 3a). In addition, *RiboJ* was inserted upstream of RFP to insulate the potential context effects of the long 5′-UTR introduced by the flanking sites[35]. For this circuit, at low pHe, int2 is repressed by PREmR34 to maintain cells in their OFF states with no fluorescence. At high pHe, PREmR34 allows int2 expression, leading to the switching of J23119 to turn cells to their ON states with high fluorescence.

Initial attempts to construct the system using wild-type T7RNAP led to a failure mode in which 100% of the cells were spontaneously switched to ON states (change in direction of J23119 leading to RFP expression) prior to induction. This result was a consequence of the leaky core pH-sensing module, which allowed a high basal level of int2 expression at pHe 5.0 (Figs. 3b and 2b, and Supplementary Fig. 8). Troubleshooting this failure mode with T7 promoter variants led to a slight recovery in the pH-sensing capacity, but > 85% spontaneous population switching at pHe 5.0 was still observed with the weakest promoter variant (Fig. 3b). The T7RNAP library provided pH-sensing capacity according to reduced polymerase processivity (Fig. 3c). Remarkably, the FQ mutant yielded a functional digital pH-sensing system with significantly reduced spontaneous switching (~ 8% population) that exhibited a digital response with an approximately 31-fold fluorescence difference between the ON (pHe 7.5) and OFF (pHe 5.0) states (Fig. 3d and Supplementary Fig. 9). A complement PCR assay show ~ 100% DNA reorientation at pHe 7.5. Thus, the switching occurred homogeneously in all RFP-bearing plasmids, and the maximal output potential of the system was achieved (Fig. 3d and Supplementary Fig. 10).

We next demonstrated that the transition band of the system, defined as the pHe range for which DNA switching is between 10 and 85%, could be flexibly adjusted by the ʟ-arabinose

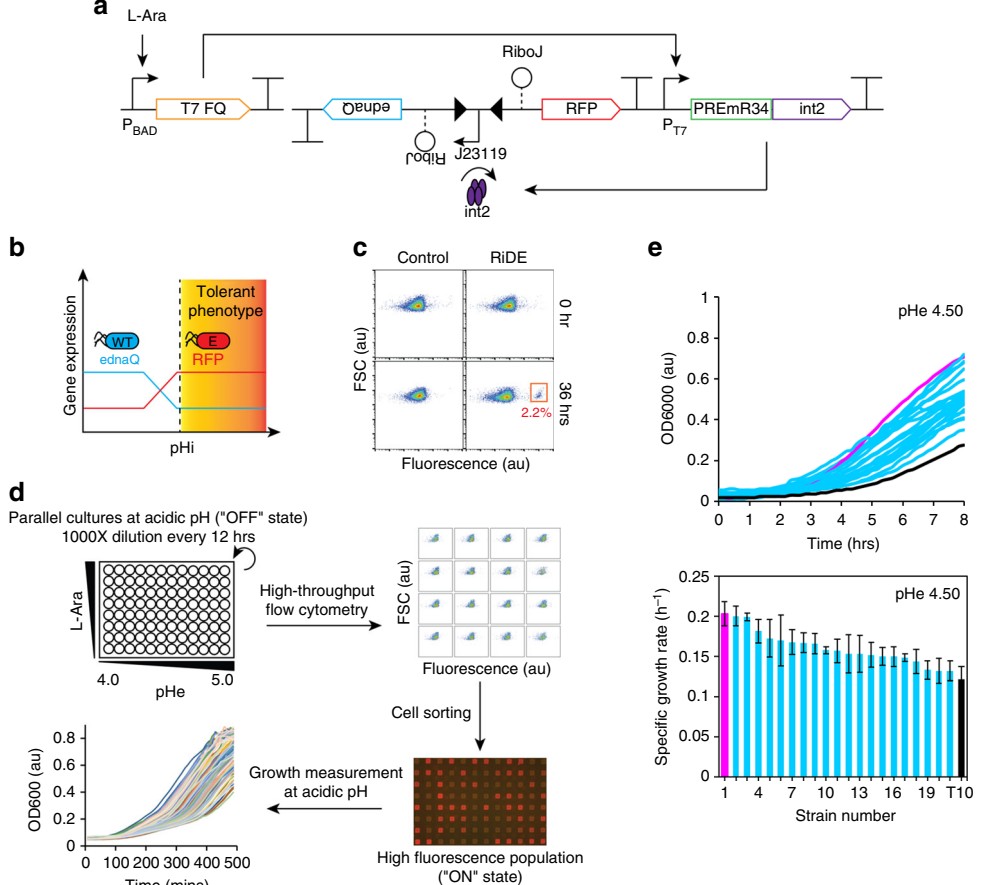

**Fig. 4** Programming autonomous evolution of acid-tolerant phenotypes. **a** Schematic of a genetic platform that enables the self-directed evolution of an acid-tolerant phenotype (RiDE). **b** Conceptual design of RiDE. A core pH-sensing module was employed to determine the single-cell pHi. In the default configuration, RiDE sets host cells to a rapidly evolving state and generates genetic heterogeneity among the progeny population. Mutants that evolve to stably maintain the pHi near neutrality activate integrase expression, which then inverts the J23119 promoter to turn OFF in vivo mutagenesis and turn ON RFP expression. **c** Flow analysis of RiDE culture after 36 h of growth. A small population of a highly fluorescent evolved phenotype is boxed in *red*. **d** High-throughput experimental workflow to isolate acid-tolerant phenotypes from RiDE-based evolution experiment. For each of 16 combinations of pHe (4.0, 4.5, 4.8, 5.0) and L-arabinose (0.0001%, 0.001%, 0.002%, 0.02% m/v), three biological replicates were prepared to form a total of 48 parallel cultures. As negative controls, cells transformed with digitalized pH-sensing system (without ednaQ expression) were used to prepare another 48 parallel cultures with similar pHe and L-arabinose conditions. RFP expressions in the parallel cultures were monitored by a robotic flow analyzer every 12 h. The highly fluorescent population was sorted, propagated, and subjected to a high-throughput growth assay at acidic pH (pHe 4.5). **e** Growth kinetics and specific growth rates of acid-tolerant strains at pHe 4.5. The data represent the mean of three biological replicates

induction level. At saturated levels of L-arabinose, the system exhibited early switching, with a transition band between pHe 5.5 and 6.5. However, when the L-arabinose inputs were decreased to reduce the expression of intracellular T7RNAP FQ, the transition band shifted horizontally toward the alkaline range by 1.0 pH unit (Fig. 3e). Conveniently, our pH-sensing device could be repurposed to sense different pH ranges by simply adjusting the chemical input channel. Finally, we determined that the memory of the recorded exposure to different pHe conditions (pHe 5.0 and 7.5) was stably maintained over prolonged periods when weakly induced with L-arabinose at a level of 0.002% m/v (Fig. 3f). This result corresponded to a switch point of approximately pHi 7.0, which was later adopted as the threshold for a directed evolution experiment in the following section of this work.

**Programmed directed evolution of acid-tolerant phenotypes.** The advanced capability to program gene expression according to the varying pHi level offers an opportunity to investigate the pH-homeostasis of biological systems. One interesting example is to

study the correlation between the pHi and cellular tolerance toward extreme acidity or alkalinity. In the microbial world, a notable characteristic contributing to the harsh pH adaptation behavior of an acidophile or alkalophile is its capacity to maintain a near-neutral cytosolic environment at extreme pHe conditions[36–39]. This ability ensures the proper function of key cellular proteins, most of which operate optimally under neutral conditions. The use of directed evolution to develop acid-adapted variants of *Saccharomyces cerevisiae* and *Lactobacillus casei* has revealed that the evolved mutants generally maintained a higher pHi compared to their parental strains grown at similar acidic pHe[40–42]. Although a consolidated perspective of all the mechanisms that contribute to cellular acid tolerance has yet to be delineated, these findings indicate that pHi is a unique indicator of this phenotype.

Using our digitalized pH-sensing device and adopting pHi as the indicator for the "acid-tolerant phenotype", we engineered a riboswitch-based genetic platform that allowed the self-directed evolution and enrichment of acid-tolerant phenotypes in the neutralophilic strain *E. coli* (RiDE) (Fig. 4a). Our design drew inspiration from an earlier study in which a mutant DNA

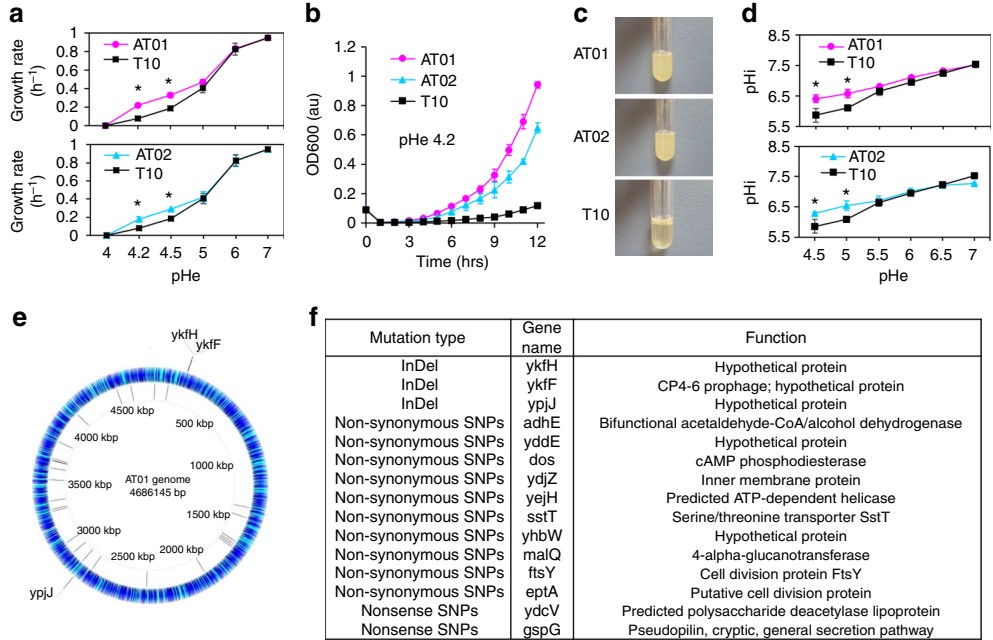

**Fig. 5** Phenotype and genotype comparison of the evolved strains AT01, AT02, and wild-type T10. **a** Growth rates of AT01, AT02, T10 at varied pHe. **b** Growth profiles of AT01, AT02, T10 in M9 medium at pHe 4.20. **c** Culture turbidity of AT01, AT02, T10 grown in M9 medium at pHe 4.20 for 18 h. **d** pHi of AT01, AT02, T10 at different pHe. Student *t*-test: *asterix* = *P* < 0.05. **e** Mutations in the AT01 genome compared to the DH10B reference. The *middle blue ring* represents coding sequences (CDSs) in the reference DH10B genome. The *arcs* in the *inner ring* represent single-nucleotide polymorphisms (*SNPs*) in AT01 compared to the reference. The *arcs* in the *outer ring* show coding sequences (ykfH, ykfF, ypjJ) with insertion or deletion mutations compared to the reference. **f** List of modified CDSs in AT01. The data represent the mean of three biological replicates

polymerase was coupled with a transcription factor-based biosensor to evolve cell factories for increased metabolite production[43]. Notably, however, we have challenged the inherent low activation ratio and sensitivity of natural riboswitches by applying various strategies to augment their sensing and output control abilities, as described above. The central concept of RiDE is to program an inverse regulation between the in vivo mutagenesis rate and RFP with respect to the pHi at the single-cell level (Fig. 4b). To introduce an in vivo mutagenesis mechanism to host cells, we employed an error-prone mutant of *dnaQ*, which is the ε subunit of *E. coli* DNA polymerase III and contributes to the proofreading capacity of the replication machinery. Intracellular expression of this *dnaQ* mutant (*ednaQ*) markedly reduced the cellular mismatch repair capacity and resulted in up to ~ $10^6$-fold increase in the error rates during genome replication, as compared to the background strain[44, 45]. Here, we observed that controlled expression of *ednaQ* led to a ~ 100-fold increase in the genome replication error rate, as compared to that of the control T10 strain (Supplementary Fig. 11). To construct RiDE, *ednaQ* was incorporated into the digitalized pH-sensing system in the reverse orientation of RFP. *RiboJ* was inserted upstream of both *ednaQ* and RFP to insulate 5′-UTR context effects. In the default architecture, RiDE constitutively expresses *ednaQ* to direct host cells into a rapid evolution state in which error-prone genome replication continuously generates genetic heterogeneity among progeny cell populations. Mutants that exhibit the capacity to buffer against changes in pHi relative to the background strain grown at a similar acidic pHe are anticipated to trigger integrase expression. Integrase expression would result in the inversion of the J23119 promoter, turning off *ednaQ* and turning on RFP expression. Since the switch is irreversible, the phenotype information is logged permanently in the form of fluorescence expression and maintained throughout the evolution experiment for assessment at arbitrary time points.

RiDE was employed to perform fluorescence-based high-throughput directed evolution and screening of acid-tolerant

*E. coli* variants (Fig. 4d). A total of 48 parallel cultures with RiDE and their corresponding controls (cells transformed with digital pH-sensing system, no *ednaQ* actuator) were incubated at varying permutations of acidic pHe (from 4.0 to 5.0) and L-arabinose concentrations. Single-cell gene expression analysis was performed on all cell cultures to evaluate their evolutionary status every 12 h using a robotic foundry with an automated sampling and flow cytometry platform. After three rounds of experimental evolution (~ 36 h), we detected a strongly fluorescent population from one RiDE culture but not in the corresponding control (Fig. 4c). We sorted this population into individual cells, propagated them, and examined their capacity to regrow in acidic media. Of the 384 tested mutants, 24 exhibited improved growth rates comparing to the background T10 at pHe 4.5 (Fig. 4e). Advantageously, the sequencing results for the plasmids revealed no or minimal sequence modifications on the RiDE plasmids except for promoter reorientation, indicating that the observed high fluorescence was due to functional RiDE switching rather than "cheating" mutations to produce false outputs of the circuit. Finally, plasmid curation through multiple rounds of sub-culturing in non-selective media was performed to rescue RiDE from the acid-tolerant strains.

**Characterization of acid-tolerant mutants.** We compared the growth properties of the two most acid-tolerant strains obtained from the self-directed evolution, AT01 and AT02, with the wild type strain T10 (Fig. 4). Overall, both AT01 and AT02 exhibited improved growth rates, as compared to T10, under acidic conditions from pHe 4.2 to 5.5 (Fig. 5a, Supplementary Fig. 12). Remarkably, prolonged exposure to pHe 4.2 inhibited the growth of T10, but not AT01 and AT02 (Fig. 5b). In fact, at this acidic pHe, AT01 grew to OD600 ~ 0.94 (~ 78% stationary phase density of culture grown at pHe 7.0) and AT02 to OD ~ 0.65 (~ 54% stationary phase density of culture grown at pHe 7.0) after

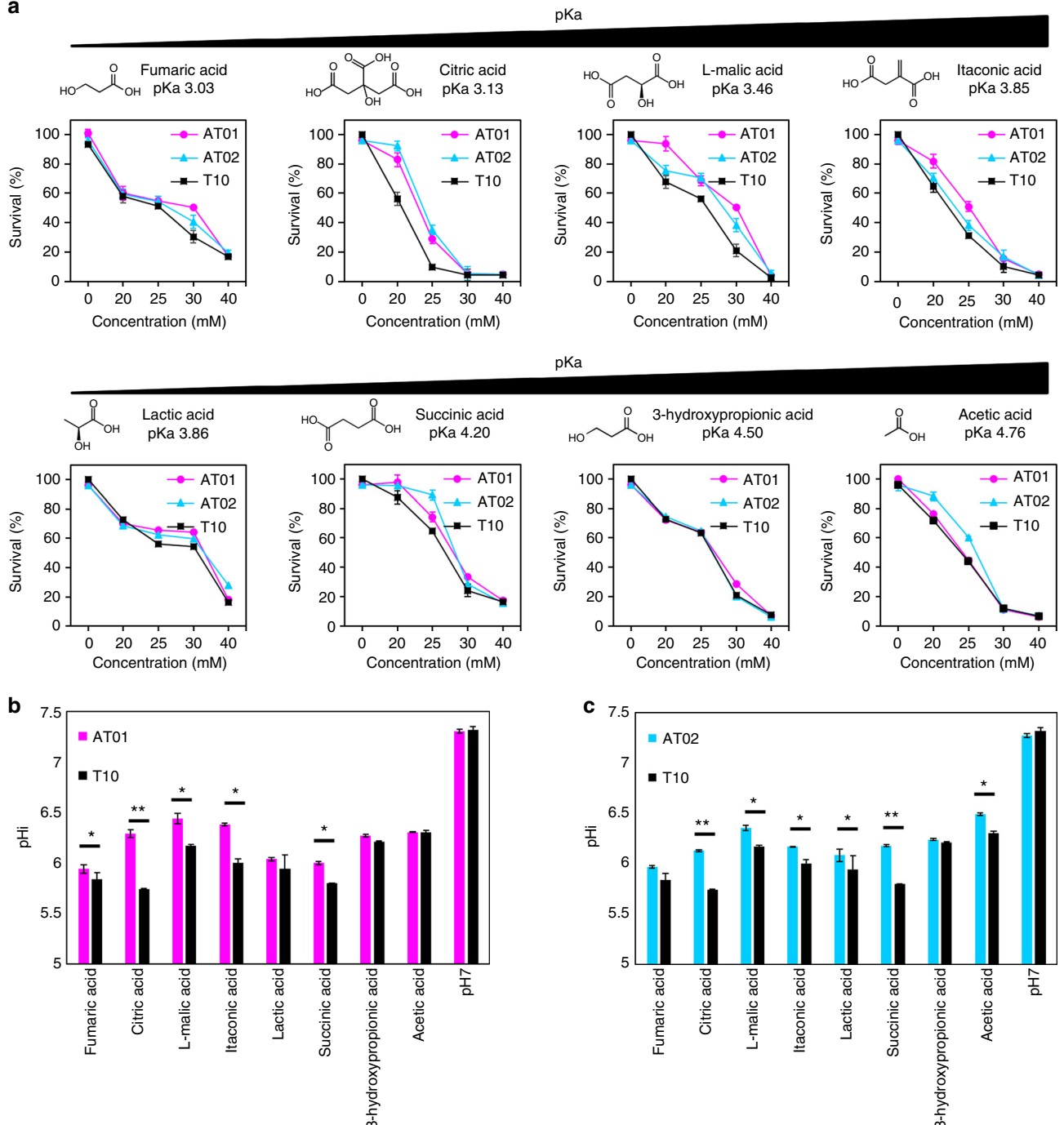

**Fig. 6** Tolerance of AT01, AT02, and T10 against industrial organic acids. **a** Survival percentage of AT01, AT02, and T10 cultures growth at varied organic acid concentrations at (37 °C, 12 h) are reported. For each organic acid, the stationery-phase cell density at each concentration were divided against maximal cell density grown in neutral media (pHe 7.0). **b** Comparison of intracellular pH (pHi) of AT01 and T10 under organic acids challenge (25 mM). **c** Comparison of intracellular pH (pHi) of AT02 and T10 under organic acids (25 mM). The pHi values are measured 6 h upon organic acid addition. Student *t*-test: *asterix* = *P* < 0.05, *double asterix* = *P* < 0.01. The data represent the mean of three biological replicates

12 h (Fig. 5b, c). We further examined the pHi of AT01, AT02, and T10 when grown under different pHe conditions ranging from 4.2 to 5.5. The results were within expectation and showed that AT01 and AT02 could maintain higher pHi levels than T10 when grown at acidic pHe, and corroborated with our initial hypothesis of adopting pHi as an indicator of acid tolerance (Fig. 5d). Specifically, AT01 maintained a 0.6 pH unit higher pHi, while AT02 maintained a 0.4 pH unit higher pHi at pHe 4.5, as compared to T10.

To further investigate the genomic modifications that underlie the acid-tolerant phenotype, we subjected AT01 and T10 to whole-genome re-sequencing, which revealed several interesting features. First, among the modifications, single-nucleotide polymorphisms (SNPs) were dominant (27 SNPs), followed by insertions and deletions (InDel) (five insertions and six deletions), and no major genome rearrangements were detected (Fig. 5e). Second, the modifications occurred randomly throughout the genome. These features were in agreement with our expectation

on the utilization of *ednaQ* as the in vivo mutagenesis mechanism. Finally, sensible modifications, including non-synonymous SNPs (resulting in amino acid mutation of coding sequences), nonsense SNPs (resulting in premature stop codons in normal coding sequences), and in-frame InDels (resulting in frameshift mutations of coding sequences), occurred in 15 coding sequences (CDSs), none of which has been associated with *E. coli* acid tolerance mechanisms prior to this study (Fig. 5f, Supplementary Table 1, and Supplementary Fig. 13). Attempts to delineate the cause for acid tolerance to specific point mutation were unsuccessful, either by complementation expression of mutant proteins in T10 or assaying for tolerant phenotype in wild type *E. coli* deletion mutants (Supplementary Fig. 14). This indicated that the acquired acid tolerance in AT01 may be an effect of multiple changes across the genome rather than isolated mutations at specific genomic locus or CDS, the exact combination of which is unknown. Future studies should critically evaluate the cause for acid tolerance by assessing the three-dimensional structural landscape of the evolved genome, variability in gene expression and metabolomic profiles of different mutants. Different permutation of SNP mutants can be generated by sequentially changing all SNP of AT01 to the original T10 through MAGE or CRISPR-Cas systems and assaying for the loss of functional phenotype[46, 47].

The industrial bioproduction of organic acids is often constrained by the limited growth capacity of the host strains in the acidic environment generated by the overproduction of their own metabolic products[48]. The development of acid-tolerant production strains in genetically tractable organisms, such as *E. coli*, therefore offers a straightforward means to improve the bioproduction yield of organic acids through increased cell mass. To explore the potential of AT01 or AT02 for this application, we compared the growth capacities of AT01, AT02, and T10 against increasing concentrations of various industrial organic acids, including fumaric acid, citric acid, L-malic acid, itaconic acid, lactic acid, succinic acid, 3-hydroxypropionic acid, and acetic acid. These organic acids were chosen based on their high industrial value, broad applications and production practicality in metabolically engineered *E. coli*[49]. Our results showed that strain AT01 exhibited improved tolerance for 6 out of the 8 tested organic acids as compared to the strain T10, except for 3-hydroxypropionic acid and acetic acid (Fig. 6a). Strain AT02 exhibited improved tolerance for 5 out of the 8 tested organic acids as compared to T10, except for lactic acid, itaconic acid and 3-hydroxypropionic acid (Fig. 6a). High-performance liquid chromatography (HPLC) analyses confirmed that the intracellular contents of each organic acid type in AT01, AT02, and T10 were comparable, suggesting that the same organic acid was equally penetrative in all three strains (Supplementary Fig. 15). To ascertain the phenotypic basis of the enhanced tolerance to organic acids, we measured the pHi of AT01, AT02, and T10 challenged with the inhibitory concentration of each organic acid by employing pHluorin2. Our results showed that the cytosolic pH (pHi) of all three strains was significantly lowered upon exposure to the organic acids, with T10 being most affected (lowest 1.1 pHi unit reduction by acetic acid, and highest 1.5 pHi unit reduction caused by citric acid). In contrast, AT01 and AT02 exhibited improved pHi buffering capacity under organic acids challenge (up to 0.5 pHi unit increase was observed for AT01 as compared to T10 under citric acid challenge) (Fig. 6b, c). The enhanced cytosolic buffering mechanisms in AT01 and AT02 likely conferred protection against cytosolic acidification from organic acids, which often led to lethal inhibition of critical intracellular enzymes. As proton stress is common in all organic acids, our results demonstrated that the

pHi-based screening mechanism of RiDE could provide a generalizable platform to rapidly engineer the tolerance of host cells to a wide range of organic acids, as shown for 7 out of 8 examples in this study. For certain weak organic acids such as 3-hydroxypropionic acid, anion toxicity might be more dominant than proton toxicity. In such a case, direct utilization of 3-hydroxypropionic acid as the selection pressure for the evolution experiment could potentially lead to the development of 3-hydroxypropionic acid tolerant strains.

## Discussion

Programming phenotypic evolution is an emerging frontier of synthetic biology[43, 50]. The intricate networks involved in phenotypic regulation represent a formidable hurdle for rational engineering attempts. This type of multi-dimensional engineering problem could instead be addressed by evolutionary engineering approaches, particularly in the modern era, when biological researchers are gaining increasing benefits from having access to a new range of ultrahigh-throughput automated platforms. The development of genetic systems that can leverage such upcoming logistic power is anticipated as a key to improving the success rate of future evolutionary engineering projects. Our study presents for the first time a generic platform that integrates a riboswitch-based controller with in vivo mutagenesis to mediate the evolutionary engineering of desirable phenotypes in *E. coli* (RiDE). In contrast to conventional adaptive evolution strategies, RiDE advantageously expedites evolution and selection in a highly autonomous and reliable manner. When implemented in a biofoundry set-up with automated sampling and FACS, RiDE effectively isolated desirable phenotypes within a few days. Such a system overcomes the challenge posed by the narrow dynamic range of engineered pH-riboswitches through a modular evolve-and-remember regime, which complements the conventional reversible sensing regime in biosensor-driven evolutionary engineering. Two major features of this new regime can be highlighted. First, this regime enables the exploitation of weak analogue sensors, such as those derived from natural riboswitches, for programmed evolutionary engineering. This aspect greatly expands the range of compatible biosensors for this engineering approach. Second, the application of memory-based DNA recording allows cell sorting to be performed at an arbitrary time point during the experiment without loss of evolved phenotypes. This capability greatly reduces the isolation of false positives due to heterogeneous gene expression, allowing progeny cells to mature and inherit stable genomic information.

## Methods

**Strains and media.** *Escherichia coli* TOP10 (Invitrogen, Singapore) (F- *mcrA* Δ (mrr-*hsd*RMS-*mcr*BC) Φ80*lacZ*ΔM15 Δ *lac*X74 *rec*A1 *ara*D139 Δ(*araleu*)7697 *gal*U *gal*K *rps*L (StrR) *end*A1 *nup*G) was used for all cloning and experiments. LB-Miller medium was used for strain propagation and maintenance. For characterization, supplemented M9 medium (1.0 mM thiamine hydrochloride, 0.2% w/v casamino acids, 0.4% glycerol, 2.0 mM MgSO$_4$, and 0.1 mM CaCl$_2$) was adjusted to the desired pH using 2.5 M HCl or 2.5 M NaOH, filter-sterilized and stored at 4 °C before use. Antibiotics were added at appropriate concentrations: kanamycin (30 μg ml$^{-1}$), ampicillin (100 μg ml$^{-1}$), and chloramphenicol (34 μg ml$^{-1}$). For inducers, concentrated stocks of L-arabinose (20% weight volume$^{-1}$ m/v) and glucose (20% m/v) were prepared in de-ionized water, filter-sterilized, and added at appropriate concentrations.

**Plasmid construction.** All plasmids were constructed following standard restriction cloning procedures and are listed in Supplementary Table 1. Q5 DNA polymerase, restriction enzymes (*Eco*RI, *Bgl*II, *Bam*HI, *Xho*I, *Aat*II), alkali phosphatase, and T4 DNA ligase were purchased from New England Biolabs (Singapore). Qiagen (Singapore) PCR purification and miniprep kits were used for DNA extraction following the manufacturer's protocols. Chemically competent *E. coli* T10 cells were prepared and transformed with appropriate constructs. Primers were purchased from a commercial synthesis service (Integrated DNA

Technologies, Singapore). Sequencing was performed for all constructed plasmids (1st Base, Singapore). Plasmid backbones are derived from standard BglBrick vectors (pBbS8a for SC101 origin, and pBbE8k for ColE1 origin). Plasmids and parts used in this study are listed in Supplementary Fig. 16, Supplementary Table 2 and 3.

**Ribosome-binding site mutagenesis.** A library of randomized RBS34 variants was constructed by performing PCR using P$_{BAD}$-RBS34-RFP as the template with the forward mutagenesis primer 5′-GGCAGATCTNNNGA GGAGNNNACTTCTATGGCGAGTAG-3′ and the reverse RFP-specific primer 5′-TGATCTCGAGTTAAGCACCGGTGGAGTGAC-3′. The PCR fragment library was digested with BglII and XhoI, ligated into pre-digested pBbE8k vector, and transformed into T10 cells. Single colonies with varying RFP expression were inoculated for plasmid miniprep and sequencing.

**Characterization of pH-sensing devices.** Single colonies of transformed cells were inoculated for 18 h at 37 °C in M9 media pH 7.0 supplemented with 0.5% m/v glucose for P$_{BAD}$ repression. The cultures were then normalized to OD600 ~ 1.0, diluted at 1:200 into 300 μl of fresh pH-adjusted M9 medium containing appropriate inducers and incubated in 96 deep-well plates for 10 h at 37 °C, 1000 rpm using Multitron shaking incubator (Infors HT, Switzerland). After induction, the cells were harvested and subjected to flow cytometry measurements.

**Characterization of digitalized pH-sensing system.** To maintain the tight regulation of the digital pH-sensing genetic circuits in the OFF state, single colonies were picked and grown in M9 medium pH 5.0 with 0.5% glucose at 30 °C, 1000 rpm for 18 h. Subsequently, stationary-phase cells were normalized to OD600 ~ 1.0 and diluted to 1:200 in 300 μl of fresh pH-adjusted M9 medium with appropriate inducer concentrations in 96 deep-well plates. These cultures were then induced for 12 h at 37 °C, 1000 rpm. Subsequently, a second round of induction was performed by diluting the cultures to 1:200 in to 300 μl of pH-adjusted M9 medium and growing for another 8 to 12 h at 37 °C, 1000 rpm using Multitron shaking incubator (Infors HT, Switzerland). The cells were then harvested and subjected to flow cytometry measurements.

**Flow cytometry measurement.** Before the measurements, cells were centrifuged at 4000×g, washed twice with cold, filter-sterilized PBS buffer and diluted to 1:400 in filtered PBS with kanamycin (2 mg ml$^{-1}$). All flow cytometry data were collected using a BD Accuri C6 Flow Cytometer (BD BioSciences, Singapore) equipped with a high-throughput C-sampler (BD Biosciences, Singapore). Data were collected at flow rate 14 μl min$^{-1}$ (10 μm core size), and bacterial cells were gated using forward scatter and side scatter. Fluorescence was detected in the FL4 channel (excitation 561 nm, detection 610/20 nm). At least 10,000 events were collected for each gated sample, and data analysis was performed with Flowjo (TreeStar, Inc., Ashland, OR). Mean fluorescence values were subtracted from white cells without RFP expression.

**High-throughput evolution for acid-tolerant phenotypes.** Escherichia coli TOP10 was co-transformed with RiDE plasmids through standard heat-shock procedure. To maintain the tight regulation of the digital pH-sensing genetic circuits in the OFF state, a single colony of transformed cells was picked and grown in M9 medium pH 5.0 with 0.5% m/v glucose at 30 °C, 1000 rpm for 18 h. Next, this stationary-phase culture was normalized to OD600 ~ 1.0. A total of 16 combinatorial conditions of pHe (4.0, 4.5, 4.8, 5.0) and L-arabinose (0.0001%, 0.001%, 0.002%, 0.02% m/v) were prepared in three replicates in a 96 deep-well plate to form 48 experimental evolution conditions. To 300 μl media of each conditions, 1.5 μl of the culture in the previous step was added. As negative controls, cells transformed with digitalized pH-sensing system (without ednaQ expression, Fig. 3a) were used to prepare another 48 parallel cultures with similar pHe and L-arabinose conditions.

For the evolution experiment, these 96 cultures were grown for 12 h at 37 °C, 1000 rpm using Multitron shaking incubators (Infors HT, Switzerland). Continuous culturing was performed every 12 h by diluting the parallel cultures 1:1000 into 300 μl of fresh M9 media with similar pHe and L-arabinose conditions. Prior to each growth cycle, all 96 parallel cultures were subjected to high-throughput flow cytometry analysis using a BD Accuri C6 Flow Cytometer (BD BioSciences, Singapore) equipped with a high-throughput C-sampler (BD Biosciences, Singapore). The negative controls showed no fluorescence expression during the evolution experiment and were used for fluorescence gating. RiDE-cultures that express RFP within this gate were subjected to further evolution. For cultures containing red cell populations (outside the fluorescence gate) indicating possible acid-tolerant phenotypes, the evolution experiment was stopped and cell sorting was performed to isolate these phenotypes for further analysis.

**Cell sorting.** Cell sorting was performed with the BD FACSJazz system (BD Biosciences, Singapore). Forward scatter and side scatter were first used for gating

of bacterial cells. From sorting culture, 50,000 events were analyzed and then a fluorescence gate was applied to sort out the high fluorescence population (fluorescence intensity > 10$^{4.5}$, excited at 488 nm and captured by a 610/20 nm filter). Single cells were sorted in drop enrich mode directly into individual wells on 96-well microtiter plates. After sorting, 200 μl of LB medium was added to each well, and the cells were inoculated at 37 °C, 1000 rpm for 18 h prior to growth characterization.

**Growth characterization.** To assess the cell growth under varied pH conditions, cells from glycerol stocks were first inoculated in 200 μl of M9 or LB medium at pH 7.0 for 18 h at 37 °C, 1000 rpm. Next, stationary-phase cultures were normalized to OD600 ~ 1.0 and diluted to 1:200 in 200 μl of fresh pH-adjusted M9 or LB medium in 96-well microtiter plates. The growth curves of these cultures were monitored at an absorbance of 600 nm for 12 h at 37 °C and 800 rpm using a Synergy HT multi-mode microplate reader (Biotek Instruments, Inc.). The maximum slope of the log-transformed blanked OD600 data at exponential growth phase (determined from six contiguous time points, 60 min) was used to determine specific growth rates.

**Organic acids tolerance assay.** Overnight stationary-phase cultures were normalized to OD600 ~ 1.0 and diluted to 1:200 into 200 μl of fresh M9 medium in 96-well microtiter plates with varied concentrations of organic acids (total volumes of concentrated organic acid stocks do not exceed 10 percent of final culture volumes). The growth curves of these cultures were recorded at an absorbance of 600 nm for 12 h at 37 °C and 800 rpm using a Synergy HT multi-mode microplate reader (Biotek Instruments, Inc.).

**Whole-genome re-sequencing.** Whole-genome re-sequencing was performed for parental Escherichia coli TOP10 and acid-tolerant strain AT01. The genomic DNA was extracted using a GenElute™ Bacterial Genomic DNA Kit (Sigma-Aldrich, Singapore). Whole-genome re-sequencing and variation analysis was performed by BGI Genomics (Hong Kong) (HiSeq2500, pair-end 125 bp, 100X coverage, data filtering). Sequencing data was mapped to reference genome Escherichia coli DH10B (taxonomy ID 316385). Genome comparison was visualized using BLAST Ring Image Generator (BRIG)[51].

**Measurement of intracellular pH.** pHluorin2 was employed for pHi measurement. Overnight cultures of cells transformed with J23119-RBS34-pHluorin2 were harvested by centrifugation at 4000×g, washed twice with PBS buffer, and normalized to OD600 ~ 1.0. Cells were then diluted to OD ~ 0.1 in 200 μl of pH-adjusted or organic acid-containing M9 medium, and grown in 96-well microtiter plates at 37 °C. The excitation spectra of the cell cultures were recorded every 2 h using a Synergy HT multi-mode microplate (Biotek Instruments, Inc.). The spectra of cultures expressing pHluorin2 were subtracted against that of blank cells (no pHluorin2 expression). The ratios of readings at wavelengths of 410 and 470 nm were calculated and used to determine cellular pHi levels following an established procedure[30].

**Characterization of error-prone DNA polymerase.** The mutator gene ednaQ was characterized by standard rifampicin reversion assays. Cells transformed with P$_{BAD}$-RBS34-ednaQ were inoculated in LB medium at 37 °C and 1000 rpm for 18 h. Subsequently, the cultures were normalized and diluted to 1:200 in 300 μl of LB medium without or with L-arabinose (0.1%) and induced at 37 °C and 1000 rpm for 12 h. After induction, the cultures were normalized to OD600 ~ 1.0, diluted to 1:10,000 and plated on LB agar containing rifampicin (50 μg ml$^{-1}$). After 18 h of incubation at 37 °C, the number of colonies on each plate was quantified and subjected to Luria-Delbruck flux analysis using the webtool Fluctuation AnaLysis CalculatOR (FALCOR) to determine the mutation rates[52, 53].

**HPLC analysis.** For HPLC analysis of intracellular organic acids accumulation, 10 ml of cell cultures after 12 h of organic acids challenge were centrifuged at 4000×g, washed twice with PBS buffer, and normalized to OD600 ~ 1.0. 1 ml of these samples were transferred to 2-ml Fastprep tubes (MP Biomedicals) with the addition of 0.1 g glass beads, acid-washed (< 106 μm, Sigma-Aldrich). Cells were lysed using a Fastprep-24 homogenizer at 6.5 ms$^{-1}$ for 60 s, repeated three times. The samples were centrifuged at 10,000 × g for 10 min, the supernatants were then collected, and filter-sterilized with a 0.22-μm filter (Sartorius Stedim) before being analyzed by an Agilent 1260 HPLC apparatus equipped with an Aminex HPX-87H Column (300 × 7.8 mm) (Bio-rad, Singapore) and a diode array detector. Compounds in the filtered culture were eluted with an isocratic pressure of 150 bars, and the mobile phase comprised an aqueous solution of 100% 4 mM H$_2$SO$_4$, and a flow rate of 0.5 ml min$^{-1}$. Detection was performed at a UV wavelength of 210 nm and a sample injection volume of 10 μl. The retention times of the samples were compared with the purified standards (Sigma-Aldrich) for identification and quantification.

**Data availability.** The genome sequence of E. coli strain AT01 has been deposited in the NCBI Sequence Read Archive and is available through GenBank (accession

no. CP022414). The authors declare that all relevant data supporting the findings of this study are available within the article and its Supplementary Information Files or from the corresponding author on request.

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

## Acknowledgements

H.L.P. is funded by the NUS Ph.D. scholarship program. M.W.C. acknowledges financial support from the Synthetic Biology Initiative of the National University of Singapore (DPRT/943/09/14), the Summit Research Program of the National University Health System (NUHSRO/2016/053/SRP/05), the Ministry of Education of Singapore (MOE/2014/T2/2/128), the US Air Force (FA/2386/12/1/4055), and the U.S. Defense

Threat Reduction Agency (HDTRA1-13-0037). This work used the resources of the NUS Bio-Foundry, a biomanufacturing research facility located at the National University of Singapore.

## Author contributions

H.L.P., A.W., and M.W.C. conceived and designed the experiments and wrote the manuscript. H.L.P. and N.C. performed the experiments. H.L.P., A.W., W.S.T., and W.S.Y. analyzed the data. M.W.C. supervised the project. All authors read and approved the final manuscript.

## Additional information

**Competing interests:** A patent application related to some of the work presented here has been filed on behalf of the National University of Singapore.

