## [Peer Review file · Nature Communications]

Reviewers' comments:

Reviewer #1 (Remarks to the Author):

In this work, the authors developed a genetic platform for evolution of acid-tolerant phenotypes. They used a previously reported pH sensor that is activated at neutral pH and repressed at low pH. This riboswitch-based pH sensor was first optimized by modifying T7 promoter strengths, T7 RNAP transcription speeds, and RBS sequences. They then implemented this optimized circuit to express integrase 2 at neutral pH. Finally, ednaQ (dnaQ mutant) was connected to the pH sensor-integrase 2 circuit to enable the self-directed evolution and enrichment of acid-tolerant phenotypes. This evolution platform was used to generate and screen strains with mutations that increase the cell's ability to maintain neutral intracellular pH.

This is an interesting engineering approach that combined previously developed genetic parts (riboswitch, T7 RNAP variants, integrase 2, and ednaQ) to develop a genetic platform for evolution of acid-tolerant phenotypes. While the approach is novel, there are some concerns as described below.

1. One main question: What would be the main benefit of their approach compared to previously demonstrated evolution approaches?

1) One benefit would be that the evolution process automatically stops when intracellular pH reaches ~7. However, wouldn't this approach limit mutant types? For example, this study did not provide mechanistic understanding of the increased acid tolerance of AT01. In addition, AT01 was tested using only extracellular acids added into the medium. Thus, AT01 might be good at minimizing effects of extracellular acids, but might be bad at dealing with intracellular acids generated from its own pathway. Would AT01 be a better strain to produce organic acids, which are generated inside the cell, than T10 if both strains are tested in neutral media? If not, can their evolution platform generate new mutants to achieve such biotechnological goals?

2) A much simpler circuit, combined with growth-based screening, would accomplish similar evolution goals. For example, a tightly-controlled inducible promoter (e.g. PBAD) can be used to express ednaQ while enriching acid-tolerant mutants, based on growth. Instead of using FACS and their complicated circuit (which may need to be removed for biotechnological applications), the same 12-h culture scheme would induce mutation when L-Ara is added and stop mutation events when L-Ara is removed. Such a simpler approach might diversify mutant types more than their approach.

2. The other concern is generalizability of their approach. In theory, their approach can be applicable to other sensor-phenotype combinations. Demonstration of their concept using other sensors would be beyond the scope of this work, but readers would want to see whether their approach can improve tolerance to 3-hydroxypropionic acid and acetic acid.

Minor points

1. Line 334-336, "Interestingly, both organic acids are characterized with pKa > 4.2, suggesting that weak organic acids may not be detrimental to E. coli growth under the tested concentration.": This statement is not true because the data clearly show growth defect at high concentrations of the mentioned acids. These acids look even more detrimental to both strains than fumaric acid, lactic acid, and succinic acid.

2. Figure 5: Are the differences in growth and pH_i between the two strains statistically significant?

3. Methods: A bit more details would be necessary for others to replicate the authors' methods.

4. Discussion: It should be revised. The novelty of this work is their approach to combine many existing genetic parts and knowledge and provide an evolution platform, as opposed to providing new pH sensors. Their discussion on designer probiotics and metabolite yield might be a bit

misleading without real demonstration.

5. Line 112, "Under neutral conditions": Should it be Under "acidic" conditions?

6. Line 115, "alkaline conditions": Should it be "neutral" conditions?

7. What is T10? Is it E. coli Top 10 strain?

8. Line 167, "between 7.2 and 8.9": pH8.9 was not tested. This should be corrected.

9. Figure 1e, S1a, S1c, S4a, S4b, S5a, S5b, S8b: The scale of the y-axis in each figure is reported incorrectly. It should be "Output x 10⁻³".

10. Figure 3b: The data for pT7m4 are missing.

11. Figure S11: It is a Table, not a figure.

12. Figure S6b: What are these values normalized to?

13. Line 195: Fig. S2c should be Fig. 2c.

14. Figure captions: More details are necessary. For example, what were L-Ara concentrations for Fig. 1e and 4?

Reviewer #2 (Remarks to the Author):

Chang and co-workers report the utilization of a pH-sensing riboswitch for controlling gene expression in E. coli. The riboswitch is optimized regarding its performance and interesting insights into the possibilities of fine-tuning of several parameters of such systems are obtained. The device is then digitalized in order to amplify responses and to create a memory effect by regulating an irreversible recombination event in the promoter region. This setup is again nicely tuned in order to perform satisfactorily. The last part of the manuscript describes the use of this setup in order to implement a self-directed evolution of acid-tolerant mutants. This was achieved by controlling the expression of an error-prone DNA polymerase via the pH-dependent recombinase system. Although the achieved acid tolerance is not very pronounced, several mutations that allow growths at decreased pH are identified by whole genome sequencing. Nevertheless, taking into account that acid tolerance seems to be a highly complex adaptation in E. coli, the obtained results are still interesting. The individual mutations are observed in genes so far not associated with acid tolerance and are not characterized further. Taken together, the work presents a very interesting and innovative way of utilizing natural riboswitch sequences in genetic engineering approaches. The experimental part is carried out carefully and state-of-the-art methods for characterizing the system have been employed. The work is described nicely, with a few exceptions that should be clarified, see below for details. In conclusion, I can recommend acceptance of this interesting work after addressing the following issues.

- It would be helpful to see how a constitutively active RFP responds to pH changes. Since pH stress will very likely have an influence on gene expression, has this effect been taken into account in the presented data?
- Eleven different recombinases have been screened for their performance regarding the promoter-switching system but the data are not shown. It should be presented in the SI.
- Some issues are not very clear due to a lack of description. For example, when describing the initial results of the recombinase approach, 8 % switching is observed. It should be mentioned that promoter recombination is meant. The same sentence then mentions 31-fold induction. This refers

to RFP fluorescence? (line238/239)

- The same applies to the description of the optimization of leakiness by RNAP engineering. At the end of the paragraph, further optimization is mentioned by tuning the RNAP concentration. However, the word "concentration" is missing in the sentence, implying that the protein sequence was optimized (line 198).
- Line 172: Literature regarding other acid-tolerant bacteria would be helpful.

We thank the reviewers for their constructive comments which helped us to improve the overall quality of the manuscript. In light of the new data in this revision, we have now revised the previous Fig. 5 to Fig. 5 and Fig. 6, respectively, in the revised manuscript. Other major changes include the revision of Supplementary Fig. S4, Supplementary Fig. S7, Supplementary Fig. S11, Supplementary Fig. S14, and Supplementary Table S1. Text changes are highlighted in red in the revised manuscript, and the figure numbering has been modified accordingly.

Reviewer #1 (Remarks to the Author):

In this work, the authors developed a genetic platform for evolution of acid-tolerant phenotypes. They used a previously reported pH sensor that is activated at neutral pH and repressed at low pH. This riboswitch-based pH sensor was first optimized by modifying T7 promoter strengths, T7 RNAP transcription speeds, and RBS sequences. They then implemented this optimized circuit to express integrase 2 at neutral pH. Finally, ednaQ (dnaQ mutant) was connected to the pH sensor-integrase 2 circuit to enable the self-directed evolution and enrichment of acid-tolerant phenotypes. This evolution platform was used to generate and screen strains with mutations that increase the cell's ability to maintain neutral intracellular pH.

This is an interesting engineering approach that combined previously developed genetic parts (riboswitch, T7 RNAP variants, integrase 2, and ednaQ) to develop a genetic platform for evolution of acid-tolerant phenotypes. While the approach is novel, there are some concerns as described below.

1. One main question: What would be the main benefit of their approach compared to previously demonstrated evolution approaches?

1) One benefit would be that the evolution process automatically stops when intracellular pH reaches ~7. However, wouldn't this approach limit mutant types?

Our genetic platform was designed to diversify host genetic information (DNA sequence) autonomously and rapidly using an error-prone DNA polymerase (ednaQ). By coupling with a feedback mechanism that permanently terminates ednaQ expression and switches to fluorescence expression (RFP) once the desired phenotypes have been acquired, the subsequent generations of the evolved phenotypes would not induce further genetic changes and instead could be isolated with cell sorting techniques. Since minimal human interference is required, our approach greatly facilitates the scale-up and automation of experimental evolution processes using modern high-throughput technologies (e.g. robotics, cell sorting).

The mutant types of interest in this specific study are novel strains that exhibit improved intracellular pH (pHi) buffering capacity to achieve better tolerance when grown in acidic conditions compared to the wild-type host. This was inspired by previous studies showing that acid-tolerant strains generally exhibit enhanced intracellular pH buffering capacity when grown in acidic environment compared to parental strains (reference 37, 40, 41). Furthermore, our overall engineering strategies and circuit designs are modular and could be easily adapted for the evolution of other interesting phenotypes, such as high producer strains exhibiting improved product accumulation compared to native strains in metabolic engineering applications. In such case, other riboswitch elements would be employed. From a more technical aspect, the generalizable engineering strategies in this study will enable the use of riboswitches with limited dynamic range for sensor-based phenotype engineering, which generally requires sensors with

wide dynamic range for accurate identification of interesting mutants. This feature greatly expands the range of phenotypes that could be engineered by the approach.

For example, this study did not provide mechanistic understanding of the increased acid tolerance of AT01.

We agree with the reviewer that the identification of specific genetic changes leading to increased acid tolerance of acquired mutants would be critical for future applications. In fact, we have performed deletion and complementation experiments on the point mutation to delineate the mechanisms of acid-tolerant property of the evolved strain AT01. However, those approaches did not yield the improvement in acid tolerance of wildtype T10 (**Supplementary Fig. S13**). From these data, we postulate that the acquired acid tolerance was mostly likely caused by multiple rather than single point mutations in AT01. In such a case, genome editing methods (MAGE, CRISPR-Cas, gene deletion and replacement) would be required to generate variants with fewer genome modifications and examine for acid tolerance. This additional study would require substantial experimentation to generate a large library of strains with combinatorial mutations on the 15 affected coding sequences (CDSs) in the genomes. In view that the main objective of this study is to demonstrate a novel memory-based directed evolution platform using riboswitches for feedback control, we believe that it is more suitable to investigate the biological basis of newly acquired acid tolerant phenotypes in another independent study.

In addition, AT01 was tested using only extracellular acids added into the medium. Thus, AT01 might be good at minimizing effects of extracellular acids, but might be bad at dealing with intracellular acids generated from its own pathway. Would AT01 be a better strain to produce organic acids, which are generated inside the cell, than T10 if both strains are tested in neutral media? If not, can their evolution platform generate new mutants to achieve such biotechnological goals?

Due to the complex production pathways of organic acids, their re-construction and testing in the evolved and wild type strains would require extensive genetic manipulation and experimentation (deletion, overexpression, and condition optimization). Rather than undertake a task that is beyond the scope of the manuscript and this revision, we have instead, in addressing the concerns of the reviewer in the revision, demonstrated to simulate high levels of engineered organic acid production; this was achieved by introducing to both the wild type T10, and the evolved strains AT01 and AT02, 25mM of organic acids (high concentration). Using HPLC and established ratiometric GFP methods (pHluorin2), we confirmed that the added organic acids were taken up, resulting in the acidification of the intracellular pH (**Supplementary Fig. S14, Fig. 6b, and 6c**). Unlike the earlier characterization with inorganic acids (**Fig. 5**), this experiment is a close approximate to metabolically engineered cells producing high levels of intracellular organic acids. Results in Fig. 6b and 6c showed that the evolved strains could confer protection against organic acid toxicity by alleviating proton stresses. A discussion of these data has now been added to the main text (**line 375-391**):

“HPLC analyses confirmed that the intracellular contents of each organic acid type in AT01, AT02 and T10 were comparable, suggesting that the same organic acid was equally penetrative in all 3 strains (Supplementary Fig. S14). To ascertain the phenotypic basis of the enhanced tolerance to organic acids, we measured the pHi of AT01, AT02, and T10 challenged with the inhibitory concentration of each organic acid by employing pHluorin2. Our results showed that

the cytosolic pH (pHi) of all 3 strains was significantly lowered upon exposure to the organic acids, with T10 being most affected (lowest 1.1 pHi unit reduction by acetic acid, and highest 1.5 pHi unit reduction caused by citric acid). In contrast, AT01 and AT02 exhibited improved pHi buffering capacity under organic acids challenge (up to 0.5 pHi unit increase was observed for AT01 as compared to T10 under citric acid challenge) (Fig. 6b and 6c). The enhanced cytosolic buffering mechanisms in AT01 and AT02 likely conferred protection against cytosolic acidification from organic acids, which often led to lethal inhibition of critical intracellular enzymes. As proton stress is common in all organic acids, our results demonstrated that the pHi-based screening mechanism of RiDE could provide a generalizable platform to rapidly engineer the tolerance of host cells to a wide range of organic acids, as shown for 7 out of 8 examples in this study.

Supplementary Figure S14. HPLC analysis of intracellular supernatants of strains AT01, AT02, T10 subjected to organic acid challenge. In all experiments, 25mM of organic acids were added to the neutral M9 media containing initial cell inoculum (OD~0.01). After 12 hours of incubation, normalized cell cultures (~OD 0.3) were extracted to harvest intracellular environment. The intracellular supernatants were analyzed by HPLC and detected at 210 nm. (a) Chromatograms show the presence of organic acids in intracellular supernatants. (b) Quantification of the

intracellular organic acid concentration of AT01 (red), AT02 (green), T10 (black). The concentrations of intracellular organic acids in 1 mL samples extracted from normalized post-acid challenge cultures (OD600~1.0) are reported. The data represent the mean of three biological replicates.

Figure 6. Tolerance of AT01, AT02, and T10 against industrial organic acids. (a) Stationary phase cell density of AT01, AT02, and T10 cultures grown at varied organic acid concentrations at 37°C for 12 hours. (b) Comparison of intracellular pH (pHi) of AT01 and T10 grown under organic acids (25 mM). (c) Comparison of intracellular pH (pHi) of AT02 and T10 grown under organic acids (25 mM). Student t-test: asterix = $P < 0.05$, double asterix = $P < 0.01$. The data represent the mean of three biological replicates.

2) A much simpler circuit, combined with growth-based screening, would accomplish similar evolution goals. For example, a tightly-controlled inducible promoter (e.g. PBAD) can be used to express ednaQ while enriching acid-tolerant mutants, based on growth. Instead of using FACS and their complicated circuit (which may need to be removed for biotechnological applications), the same 12-h culture scheme would induce mutation when L-Ara is added and stop mutation events when L-Ara is removed. Such a simpler approach might diversify mutant types more than their approach.

We agree with the reviewer that a simpler approach combining expression of ednaQ in host cells with growth-based screening could theoretically accomplish similar evolution goals. Nonetheless, without a feedback mechanism, error-prone genome replication would continue relentlessly until L-ara removal. Given the high genome error rates generated by ednaQ, this would lead to a large number of additional mutations in evolved strains that do not contribute to the desired phenotypes. In addition, the undesired mutations may lead to the development of growth defects, antibiotic resistance, and spontaneous DNA damage in evolved phenotypes, thus reducing the overall success rate of the evolution experiments.

Our genetic platform enables the directed evolution experiment to be stopped in a timely fashion once the phenotypes have been acquired. The genomic information (DNA sequence) of subsequent generations of evolved phenotypes in the cultures will therefore remain stable for further assessment. In addition, the added advantage that the evolved cells would constitutively express fluorescence as soon as ednaQ function is terminated allows the evolved cells to be easily isolated by FACS sorting at a convenient time point. Taken together, the platform allows us to set up directed evolution experiments with minimal human interference (manual isolation, inducer addition and removal), and high degree of automation using high-throughput technologies (e.g. robotics, cell sorting).

2. The other concern is generalizability of their approach. In theory, their approach can be applicable to other sensor-phenotype combinations. Demonstration of their concept using other sensors would be beyond the scope of this work, but readers would want to see whether their approach can improve tolerance to 3-hydroxypropionic acid and acetic acid.

We agree with the reviewer on the comments. In this revision, we characterized other evolved strains in the mutant library (**Fig. 4**), in addition to AT01, for tolerance towards 3-hydroxypropionic acid (3-HP) and acetic acid. Three strains including AT02 exhibited improved tolerance towards acetic acid. In the revised manuscript, we report the survival percentages, intracellular pHs, growth profiles of the strain AT02 in the presence of organic acids (**Fig. 5, Fig. 6, and Supplementary Fig. S11**). AT01 and AT02 collectively exhibited improved tolerance for 7 out of the 8 tested organic acids (except for 3-hydroxypropionic acid). A discussion on this new data set is now included in the main text (**line 327-339 and line 362-375**):

“We compared the growth properties of the two most acid-tolerant strains obtained from the self-directed evolution, AT01 and AT02, with the wild type strain T10 (Fig. 4). Overall, both AT01 and AT02 exhibited improved growth rates, as compared to T10, under acidic conditions from pHe 4.2 to 5.5 (Fig. 5a, Supplementary Fig. S11). Remarkably, prolonged exposure to pHe 4.2 inhibited the growth of T10, but not AT01 and AT02 (Fig. 5b). In fact, at this acidic pHe, AT01 grew to OD₆₀₀~0.94 (~78% stationary phase density of culture grown at pHe 7.0) and AT02 to OD~0.65 (~54% stationary phase density of culture grown at pHe 7.0) after 12 hours (Fig. 5b, Fig. 5c). We further examined the pH_i of AT01, AT02, and T10 when grown under different pHe

conditions ranging from 4.2 to 5.5. The results were within expectation and showed that AT01 and AT02 could maintain higher pHi levels than T10 when grown at acidic pHe, and corroborated with our initial hypothesis of adopting pHi as an indicator of acid tolerance (Fig. 5d). Specifically, AT01 maintained a 0.6 pH unit higher pHi, while AT02 maintained a 0.4 pH unit higher pHi at pHe 4.5, as compared to T10.”

“The industrial bioproduction of organic acids is often constrained by the limited growth capacity of the host strains in the acidic environment generated by the overproduction of their own metabolic products⁴⁸. The development of acid-tolerant production strains in genetically tractable organisms, such as *E. coli*, therefore offers a straightforward means to improve the bioproduction yield of organic acids through increased cell mass. To explore the potential of AT01 or AT02 for this application, we compared the growth capacities of AT01, AT02, and T10 against increasing concentrations of various industrial organic acids, including fumaric acid, citric acid, L-malic acid, itaconic acid, lactic acid, succinic acid, 3-hydroxypropionic acid, and acetic acid. These organic acids were chosen based on their high industrial value, broad applications and production practicality in metabolically engineered *E. coli*⁴⁹. Our results showed that strain AT01 exhibited improved tolerance for 6 out of the 8 tested organic acids as compared to the strain T10, except for 3-hydroxypropionic acid and acetic acid (Fig. 6a). Strain AT02 exhibited improved tolerance for 5 out of the 8 tested organic acids as compared to T10, except for lactic acid, itaconic acid and 3-hydroxypropionic acid (Fig. 6a).”

Both proton and anion components contribute to organic acids toxicity in cells. As proton stress is common in all organic acids, the pHi-based screening mechanism of RiDE provide a generalizable platform to rapidly engineer tolerant phenotypes to a wide range of organic acids, as shown for 7 out of 8 organic acids in this study. For a specific case of weak organic acid 3-hydroxypropionic acid, we hypothesize that the toxicity of anion components is more dominant and cannot be overcome by cytosolic neutralization. This discussion has now been included in the main text (line 391-394):

“For certain weak organic acids such as 3-hydroxypropionic acid, anion toxicity might be more dominant than proton toxicity. In such a case, direct utilization of 3-hydroxypropionic acid as the selection pressure for the evolution experiment could potentially lead to the development of 3-hydroxypropionic acid tolerant strains.”

Figure 5. Phenotype and genotype comparison of the evolved strains AT01, AT02, and wild-type T10. **(a)** Growth rates of AT01, AT02, T10. **(b)** pH_i of AT01, AT02, T10 at different pH_e . Student t-test: asterix = $P < 0.05$. **(c)** Growth profiles of AT01, AT02, T10 in M9 medium at pH_e 4.20 **(d)** Culture turbidity of AT01, AT02, T10 grown in M9 medium at pH_e 4.20 for 18 hours. **(e)** Mutations in the AT01 genome compared to the DH10B reference. The middle blue ring represents coding sequences (CDSs) in the reference DH10B genome. The arcs in the inner ring represent single-nucleotide polymorphisms (SNPs) in AT01 compared to the reference. The arcs in the outer ring show coding sequences (ykfH, ykfF, ypjJ) with insertion or deletion mutations compared to the reference. **(f)** List of modified CDSs in AT01. The data represent the mean of three biological replicates.

Supplementary Figure S11. Growth profiles of AT01, AT02, and T10 at varying pH_e levels. Data represents mean of three biological replicates.

Minor points

1. Line 334-336, “Interestingly, both organic acids are characterized with $pK_a > 4.2$, suggesting that weak organic acids may not be detrimental to *E. coli* growth under the tested concentration.” This statement is not true because the data clearly show growth defect at high concentrations of the mentioned acids. These acids look even more detrimental to both strains than fumaric acid, lactic acid, and succinic acid.

We agree with the reviewer on this incorrect interpretation and have removed the sentence from the discussion.

2. Fig. 5: Are the differences in growth and pH_i between the two strains statistically significant?

We performed student t-tests to evaluate the differences in growth and pH_i between the strains. **Fig. 5a** showed that the growth rates of AT01 and AT02 at acidic pH_e (4.5 and 5.0) were significantly different from T10 ($p < 0.05$). **Fig. 5d** showed that pH_i of AT01 and AT02 at both pH_e 4.5 and pH_e 5.0 were significantly higher than T10 ($p < 0.05$). The significance was also indicated in **Fig. 6b and 6c** respectively.

3. Methods: A bit more details would be necessary for others to replicate the authors’ methods.

The methods for “High-throughput directed evolution for acid-tolerant phenotypes” and “Measurement of intracellular pH” have been revised for better clarity. The methods for “HPLC analysis” have now been added.

4. Discussion: It should be revised. The novelty of this work is their approach to combine many existing genetic parts and knowledge and provide an evolution platform, as opposed to providing new pH sensors. Their discussion on designer probiotics and metabolite yield might be a bit misleading without real demonstration.

The discussion section was revised as suggested. The first paragraph has now been removed for a more focused discussion on the major novelty of the study.

5. Line 112, “Under neutral conditions”: Should it be Under “acidic” conditions?

We have changed “Under neutral conditions” to “Under extracellular pH (pH_e) 6.8”. The test condition is mentioned based on a previous study (reference 27). This sentence is now in **line 114**.

6. Line 115, “alkaline conditions”: Should it be “neutral” conditions?

We have changed “alkaline conditions” to “ pH_e 8.0”. The test condition is now mentioned based on a previous study (reference 27). This sentence is now in **line 117**.

7. What is T10? Is it *E. coli* Top 10 strain?

Yes. T10 is *E. coli* TOP10 (see **line 166**).

8. Line 167, “between 7.2 and 8.9”: $pH_{8.9}$ was not tested. This should be corrected.

The values in this statement were intracellular pH (pHi) derived from Supplementary Fig. S2. These values have now been corrected to “between 7.2 and 8.3” in line 169 – 170. Supplementary Fig. S2 and S5 have been cited to enhance the clarity of the manuscript.

9. Fig. 1e, S1a, S1c, S4a, S4b, S5a, S5b, S8b: The scale of the y-axis in each Fig. is reported incorrectly. It should be “Output x 10⁻³”.

We have corrected the scale of y-axes in Fig. 1e, S1a, S1c, S4a, S4b, S5a, S5b, S9b (previously S8b).

10. Fig. 3b: The data for pT7m4 are missing.

We thank the reviewers for pointing out this critical omission. The legend and data for Fig. 3b have been revised. Kindly note that the graphs of pT7 and pT7m1 overlapped for most coordinates.

11. Fig. S11: It is a Table, not a Figure.

We have changed the previous “Supplementary Fig. S11” to “Table S1” in the revised manuscript.

12. Fig. S6b: What are these values normalized to?

For clarity, the y-axis has been re-labeled as RFP/A600, representing population based fluorescence reading normalized to cell optical density at 600nm.

13. Line 195: Fig. S2c should be Fig. 2c.

This has been corrected. The sentence is now in line 205.

14. Fig. captions: More details are necessary. For example, what were L-Ara concentrations for Fig. 1e and 4?

The legends of Fig. 1e and Fig. 4 have been revised.

Reviewer #2 (Remarks to the Author):

Chang and co-workers report the utilization of a pH-sensing riboswitch for controlling gene expression in *E. coli*. The riboswitch is optimized regarding its performance and interesting insights into the possibilities of fine-tuning of several parameters of such systems are obtained. The device is then digitalized in order to amplify responses and to create a memory effect by regulating an irreversible recombination event in the promoter region. This setup is again nicely tuned in order to perform satisfactorily. The last part of the manuscript describes the use of this setup in order to implement a self-directed evolution of acid-tolerant mutants. This was achieved by controlling the expression of an error-prone DNA polymerase via the pH-dependent recombinase system. Although the achieved acid tolerance is not very pronounced, several mutations that allow growths at decreased pH are identified by whole genome sequencing. Nevertheless, taking into account that acid tolerance seems to be a highly complex adaptation in *E. coli*, the obtained results are still interesting. The individual mutations are observed in genes so far not associated with acid tolerance and are not characterized further. Taken together, the work presents a very interesting and innovative way of utilizing natural riboswitch sequences in genetic engineering approaches. The experimental part is carried out carefully and state-of-the-art methods for characterizing the system have been employed. The work is described nicely, with a few exceptions that should be clarified, see below for details. In conclusion, I can recommend acceptance of this interesting work after addressing the following issues.

- It would be helpful to see how a constitutively active RFP responds to pH changes. Since pH stress will very likely have an influence on gene expression, has this effect been taken into account in the presented data?

This effect has been examined in greater detail and reported in Supplementary Fig. S4. Further interpretation has now been added in the main text (line 176-181):

“Furthermore, the effects of pHe on the well-known promoters used in this study, P_{BAD} and J23119, were investigated (Supplementary Fig. S4). For both promoters, at pHe 5.0, a slight reduction in RFP expression was observed (~8% of maximal expression at pHe 7.0). Above this pHe level, RFP expression remained constant. Taking into consideration that the sensitive range of the pH-sensor is between pHe 6.0 to 8.0, the pH-dependent gene expression can be interpreted to be under the regulation of the riboswitch element.”

Supplementary Fig. S4. Effect of pHe on promoter activities of pBAD and J23119. (a) Left: representative flow cytometry at pHe 5.0 (blue) and pHe 8.0 (red) of P_{BAD} promoter. Right: output from P_{BAD} at varied pHe levels (LA 0.01%). (b) Left: representative flow cytometry at pHe 5.0 (blue) and pHe 8.0 (red) of J23119 promoter. Right: output from J23119 at varying pHe levels. Data represents mean of three biological replicates.

- Eleven different recombinases have been screened for their performance regarding the promoter-switching system but the data are not shown. It should be presented in the SI.

We thank the reviewer for the suggestion. The behavior of this recombinase library has already been extensively characterized in a previous study (reference 34). In the current study, we selected and validated the performance of candidate recombinases with a wide dynamic range and tolerance toward expression leakiness (int2, int4, int5, int8, int9). The data are included as Supplementary Fig. S7. Further description has been added to the main text (line 228-235):

“Previously, it was shown that the expression of these integrases above a critical threshold within host cells induces irreversible DNA switching with no cross-talk to the *E. coli* recombinase system. We selected 5 integrases (int2, int4, int5, int8, int9) from this library and compared their dynamic behavior in a 1-plasmid system within a configuration relevant to our downstream applications (Supplementary Fig. S7a). In agreement with previous studies, integrase 2 (int2) exhibited a wide dynamic range and high tolerance toward leaky expression levels (Supplementary Fig. S7b). This integrase was therefore used for further circuit construction.”

Supplementary Fig. S7. Comparison of the dynamic behavior of integrase library. **(a)** Schematic of genetic constructs used to compare the dynamic behavior of integrase library. **(b)** Dynamic behavior of integrase library. The 1-plasmid system was induced with varied L-arabinose levels, and the population of “ON” state cells were measured. The data represents means of three independent experiments.

- Some issues are not very clear due to a lack of description. For example, when describing the initial results of the recombinase approach, 8% switching is observed. It should be mentioned that promoter recombination is meant. The same sentence then mentions 31-fold induction. This refers to RFP fluorescence? (line238/239)

This sentence in line 244-245 has been revised:

“100% of the cells were spontaneously switched to ON states (change in direction of J23119 leading to RFP expression) prior to induction.”

The 31-fold induction refers to RFP expression. The sentence in line 251-254 has been revised:

“Remarkably, the FQ mutant yielded a functional digital pH-sensing system with significantly reduced spontaneous switching (~8% population) that exhibited a digital response with an approximately 31-fold fluorescence difference between the ON (pHe 7.5) and OFF (pHe 5.0) states (Fig. 3d and Supplementary Fig. S9).”

- The same applies to the description of the optimization of leakiness by RNAP engineering. At the end of the paragraph, further optimization is mentioned by tuning the RNAP concentration. However, the word “concentration” is missing in the sentence, implying that the protein sequence was optimized (line 198).

This has been corrected. The sentence is now in line 207-210:

“Further optimization of the intracellular T7RNAP concentrations by controlling L-arabinose induction levels ultimately yielded a pH-sensing device with significantly reduced background expression in the OFF state (output 214 rfu) (Fig. 2c, Fig. 2d, and Fig. 2e).”

- Line 172: Literature regarding other acid-tolerant bacteria would be helpful.

Additional literature has been added as reference 31.

REVIEWERS' COMMENTS:

Reviewer #1 (Remarks to the Author):

The points raised in the previous review have been addressed.